# Probiotics as a Tool for Regulating Molecular Mechanisms in Depression: A Systematic Review and Meta-Analysis of Randomized Clinical Trials

**DOI:** 10.3390/ijms24043081

**Published:** 2023-02-04

**Authors:** Michalina Sikorska, Anna Z. Antosik-Wójcińska, Monika Dominiak

**Affiliations:** 1Medical Center of Postgraduate Education, Medical University of Warsaw, Żwirki i Wigury 61, 02-091 Warsaw, Poland; 2Department of Psychiatry, Faculty of Medicine, Collegium Medicum, Cardinal Wyszynski University in Warsaw, Woycickiego 1/3, 01-938 Warsaw, Poland; 3Department of Pharmacology, Institute of Psychiatry and Neurology, Sobieskiego 9, 02-957 Warsaw, Poland

**Keywords:** depression, microbiota, probiotics, mood disorders, gut-brain axis, psychobiotics

## Abstract

Depression is one of the main mental disorders. Pharmacological treatment of depression is often associated with delayed effects or insufficient efficacy. Consequently, there is a need to discover new therapeutic methods to cope with depression faster and more effectively. Several lines of evidence indicate that the use of probiotic therapy reduces depressive symptoms. Nonetheless, the exact mechanisms linking the gut microbiota and the central nervous system, as well as the potential mechanisms of action for probiotics, are still not entirely clarified. The aim of this review was to systematically summarize the available knowledge according to PRISMA guidelines on the molecular mechanisms linking probiotics and healthy populations with subclinical depression or anxiety symptoms, as well as depressed patients with or without comorbid somatic illnesses. The standardized mean difference (SMD) with 95% confidence intervals (CI) was calculated. Twenty records were included. It has been found that probiotic administration is linked to a significant increase in BDNF levels during probiotic treatment compared to the placebo (SMD = 0.37, 95% CI [0.07, 0.68], *p* = 0.02) when considering the resolution of depressive symptoms in depressed patients with or without comorbid somatic illnesses. CRP levels were significantly lower (SMD = −0.47, 95% CI [0.75, −0.19], *p* = 0.001), and nitric oxide levels were significantly higher (SMD = 0.97, 95% CI [0.58, 1.36], *p* < 0.0001) in probiotic-treated patients compared to the placebo, however, only among depressed patients with somatic co-morbidities. There were no significant differences in IL-1β, IL-6, IL-10, TNF-α, and cortisol levels after probiotic administration between the intervention and control groups (all *p* > 0.05). Firm conclusions on the effectiveness of probiotics and their possible association with inflammatory markers in the healthy population (only with subclinical depressive or anxiety symptoms) cannot be drawn. The advent of clinical trials examining the long-term administration of probiotics could evaluate the long-term effectiveness of probiotics in treating depression and preventing its recurrence.

## 1. Introduction

Depression is a main mental disorder. Since current treatments for depression unfortunately still have a high failure rate, they pose a serious medical challenge. Hence, there is an urgent need to search for new therapies. The search for both new antidepressant molecules and non-pharmacological therapies is ongoing. For nearly a decade, increasing attention has been paid to the need for a holistic approach to health, especially for mental disorders. This has resulted in an interesting line of research relating to gut microbiota. Studies have shown that there are significant differences in the composition and abundance of gut microbiota in healthy individuals and those suffering from depression [1]. Interestingly, the above thesis is supported by studies conducted on animal models. After transplanting gut microbiota from patients suffering from depression, altered animal behavior corresponding to depression was observed in rodents. Hence, it was hypothesized that this might be related to the influence of the gut microbiome on the metabolism [2]. Further research has focused on a method of reprogramming the gut microbiome to create a composition that is more similar to healthy patients. One of the simplest and least invasive methods is reprogramming the gut microbiome with probiotics [3].

Probiotics are living microorganisms developed through biotechnology that adapts to the human gut and provides health benefits. To date, they have been proven to be effective in a number of conditions, such as acute diarrhea, various types of allergies and inflammatory diseases, including atopic dermatitis, and inflammatory bowel disease [4,5]. Relevant to these cases is the concept of intestinal dysbiosis, which is a disruption in the composition and quantity of the intestinal microbiota as a result of chronic inflammation, which is observed in depression [6]. Clinical studies that have been conducted to date have been summarized in a meta-analysis of randomized controlled trials (RCTs) evaluating the effects of probiotics on depressive symptoms. This meta-analysis showed that probiotics could benefit patients suffering from depression [7].

Studies focusing on determining the importance of probiotics in depression have also reported mitigating inflammation by lowering the circulating levels of pro-inflammatory markers [8].

There are at least several molecular mechanisms that are relevant to the action of probiotics, which are summarized in this review. The most important benefits of probiotic-regulated molecular mechanisms include an improvement in the intestinal barrier function and tightness, the regulation of gut-associated lymphoid tissue (GALT) functions, the increased synthesis of anti-inflammatory cytokines, the regulation of vagus nerve activity, the regulation of γ-aminobutyric acid (GABA) binding receptor expression, the regulation of the hypothalamic-pituitary-adrenal axis, and reduction in cortisol: a hormone secreted during chronic stress. There are many reviews in the literature on the effectiveness of probiotics in depressive disorders [9,10,11,12]; however, to our knowledge, there is no literature review on the molecular mechanisms underlying the effects of microbiota dysregulation and probiotic therapy on depressive disorders.

The purpose of this review was to present the current state of knowledge on the molecular mechanisms involved in the gut-brain axis in three groups of patients treated with probiotics: (1) healthy participants with subclinical depressive or anxiety symptoms, (2) depressed patients, and (3) patients with depression and comorbid somatic illness. We hypothesized that the treatment results and molecular mechanisms might vary according to the severity of the depressive symptoms, as well as the presence of somatic disorders. To our knowledge, such a review has not been conducted to date.

## 2. Methods 

On 30 April 2022, a search for articles in MEDLINE, Embase, and PsycINFO was conducted, and on 1 November 2022, a search in the same databases was updated. We used terms targeting gut microbiota, probiotic therapy, and depression during the search. The search used medical subject headings (MeSH), text words, and keywords. The following combinations of words were used: (‘gut-brain axis’ OR ‘gut microbiota’ OR ‘intestinal microbiota’ OR ‘probiotics’ OR ‘psychobiotics’ OR ‘Lactobacillus’ OR ‘Bifidobacterium’ OR ‘prebiotics’) AND (‘depression’ OR ‘major depression’ OR ‘mood disorders’). The results were structured as proposed by the preferred reporting items for systematic reviews and meta-analyses (PRISMA) (figure below). At first, publications relating to clinical trials published before 2015 were rejected to provide the most recent data. In addition, the deduplication of results was performed. The next step was the exclusion of studies conducted on an animal model, book chapters, review articles, mini-reviews, encyclopedias, conference abstracts, correspondence, short communications, discussions, editorials, letters, notes, and short surveys. The search included only publications in the English language. After analyzing the abstracts of the retrieved articles, studies that analyzed the molecular mechanism of action during administration in patients suffering from depression were extracted. Other publications which did not meet the above conditions, including non-original articles, articles without the full text available, in a different language than English, or not related to PICO questions (Table 1), were excluded. Other criteria of exclusion were studies not describing patients with depression or depressive symptoms and the molecular impact of probiotics on these conditions. The literature was searched, selected, and identified independently by two researchers (MS, MD). Subsequently, reviewers (MS, MD) independently identified potentially relevant articles and reviewed the titles and abstracts. To access eligibility criteria, the full text of all qualified or tentative studies was obtained. Finally, all disagreements between reviewers were resolved through discussions to reach a consensus. Our review considered the following information contained in the scientific items presented: authors, year of publication, country, study design, study and control sample size, study duration, study group characteristics (gender, mean age, diagnosis, treatment), type of probiotic, results based on professional scales assessing the incidence of depression, described in detail in the Section 3.

The meta-analysis was performed using Review Manager (RevMan) version 5.3. Only parameters assessed in at least two good-quality studies (low/moderate risk of bias) were included in the meta-analysis, with the remaining results reported in descriptive analyses. Studies with sufficient data on the mean differences in the levels of IL-1β, IL-6, IL-10, TNF-α, CRP, BDNF, cortisol, lipids and nitric oxide and oxidative stress along with standard deviations (SD) for both the intervention (probiotic) and control (placebo) groups were considered for the meta-analysis. The standardized mean difference (SMD) and its 95% confidence intervals (CI) for the effect of probiotics on serum levels of the respective markers in the intervention and control groups were estimated. Heterogeneity between studies was measured using Tau-square, chi-square, and I-square statistics; an I-square ≥50% was considered to indicate significant heterogeneity. In order to estimate publication bias, funnel plots of precision were evaluated. Subgroup analyses to assess whether results were influenced by the group (healthy subjects/patients with depression without comorbidities/patients with depression with somatic comorbidities) were not feasible due to the lack of studies in the respective subgroups.

## 3. Results and Discussion

### 3.1. Study Characteristics

Within the 5819 citations obtained, 2714 did not meet the inclusion criteria and were excluded before the screening. Among the remaining 2999 articles, 2714 were removed since they constituted duplicates in the searched databases. Following a check of titles and abstracts, 2429 publications were excluded as they involved animal models in studies, had a review paper nature, or prospective observational design nature. However, 20 records met the inclusion criteria. The results are shown in Figure 1.

The search strategy identified 5819 abstracts. We reviewed 285 publications found in databases on the use of probiotics in depressed patients. Twenty clinical trials met the inclusion criteria (Table 2, Table 3 and Table 4). Two of these papers were identified as the same clinical study; thus, data from this study were interpreted together, resulting in a total of 19 clinical trials that were evaluated in this review. Seventeen of them were double-blind randomized placebo-controlled trials, and two of them were open-label RCTs. Patients from all studies comprised a group of 1461, with a group size ranging from 11 to 156 participants. Most of the studies were performed in Iran (7), followed by Austria (2) and China (2), with single representations from Russia (1), Turkey (1), Korea (1), Poland (1), New Zealand (1), Taiwan (1), Switzerland (1), and India (1).

The participants involved in the research had identified depressive symptoms or were diagnosed with depression. Participants were screened and examined according to the Diagnostic and Statistical Manual of Mental Disorders (DSM) criteria, using the Hamilton Rating Scale for Depression (HAM-D), Beck Depression Inventory (BDI), Beck Depression Inventory-II (BDI-II), Hospital Anxiety and Depression Scale (HADS), Beck Anxiety Inventory (BAI), Montgomery–Asberg Depression Rating Scale (MADRS), 11-item Centre for Epidemiological Studies–Depression Scale (CES-D), Symptom Checklist-Revised (SCL-90R), Positive and Negative Syndrome Scale (PANSS), Quality of Life Scale (QOLS), Pittsburgh Sleep Quality Index (PSQI), Spielberger state-trait anxiety inventory form Y (STAI-Y), International Physical Activity Questionnaire (IPAQ), and Mania Self Rating Scale (MSS) [14,15,16,17,18,19,20,21,22,23,24,25,26,27,28,29,30,31,32]. 

In two research papers, patients were suffering from subclinical depressive or anxiety symptoms (Table 2) [21,25], while in 10 trials, the study group consisted of patients with major depressive disorder (Table 3) [14,15,18,20,22,24,25,27,28,29]. In seven studies, patients were diagnosed with somatic diseases, including one study with hemodialysis patients, one study with irritable bowel syndrome, one with constipation, one study with a recent diagnosis of myocardial infarction, and three studies involving patients with coronary artery disease (Table 4) [17,19,26,30,31,32,33]. Regarding probiotics, they were administered as a single-strain in seven studies [25,26,28,29,31,32,33], two strains in three studies [16,20,23], three strains in one study [24], four strains in two studies [21,30], seven strains in seven studies [17,19], nine strains in three studies [15,22,27], or fourteen strains of bacteria in one study [18]. The strains used in probiotic administrations belonged to the *Bacillus, Bifidobacterium, Lacticaseibacillus, Lactobacillus, Lactococcus,* and *Streptococcus* genus. The duration of the study ranged from 4 to 12 weeks. No serious adverse effects were reported in any of the studies described.

### 3.2. Quality Assessment

Interventional studies that were included in the review were assessed according to the RoB 2 tool (a revised tool for assessing the risk of bias in randomized trials) [34,35] and the ROBINS-I tool (risk of bias in non-randomized studies-of interventions) [36]. Among eighteen interventional studies, four were ranked as “low risk of bias” [14,17,19,21], nine were ranked as “some concerns” [15,20,22,23,28,30,31,32,33], and five were ranked as “high risk” [18,24,26,27,29]. One open trial study was evaluated using the ROBINS-I tool and was assessed as a “serious risk” [25]. The risk of bias for all the studies is presented in Figure 2 and Figure 3.

**Table 2 ijms-24-03081-t002:** A Collection of clinical studies on the effects of probiotic supplementation and molecular mechanisms of its action in healthy participants with depressive or anxiety symptoms.

Author, Year, Country	Study Design	Characteristics of the Research Group	Sample Size (N), Sex, Average Age	Intervention	Study Methodology: Measurements of Effects, Duration of the Intervention	Conclusions	Molecular Mechanism of Action
Lee et al., 2021, Korea [21]	Double-blind, randomized placebo- controlled trial	156 healthy participants with subclinical symptoms of depression, anxiety, and insomnia	156, M, F, Average age 38 years old	*Lactobacillus reuteri NK33* and *Bifidobacterium* *adolescentis NK98* (2.0 × 109 CFU for *Lactobacillus reuteri NK33* and 0.5 × 109 CFU for *Bifidobacterium adolescentis NK98*) 2 capsules per day	SRI, BDI-II, BAI, PSQI, ISI, IL-6, feces bacterial genomic DNA examination Duration—8 weeks	Significant improvement in depressive symptoms, sleep quality, and anxiety symptoms probiotic treatment led to a decrease in serum interleukin-6 levels	Assessing the inflammatory response and hypothalamic-pituitary-adrenal axis activity during probiotic NVP-1704 administration
Romijn et al., 2017, New Zeland [23]	Double-blind, randomized placebo- controlled trial	79 patients not currently taking psychotropic medications with at least moderate scores on self-reported mood measurements and symptoms	79, M, F, Average age 35 years old	*Lactobacillus helveticus R0052* and *Bifidobacterium longum R0174* 1.5 g sachet per day	MADRS, IBS-SSS, hs-CRP, IL-1β, IL-6, TNF-α, Vitamin D, BDNF; Duration—8 weeks	No results showing improvement in patients’ mood after 8-week treatment; No significant difference was found between the probiotic and placebo groups on any blood-based biomarker	Evaluation of the level of blood inflammatory biomarkers, vitamin D, and brain-derived neurotrophic factor and their impact or prediction on treatment response

F—female; M—male; CFU—Colony Forming Unit; SRI—Stress Response Inventory; BDI-II—Beck Depression Inventory-II; BAI—Beck Anxiety Inventory; PSQI—Pittsburgh Sleep Quality Index; ISI—Insomnia Severity Index; IL-6—Interleukin-6; DNA—deoxyribonucleic acid; MADRS—Montgomery and Asberg Depression Rating Scale; IBS-SSS—Irritable Bowel Syndrome Symptom Severity Scale; hs-CRP—High-Sensitivity CRP; IL-1β—Interleukin-1β; TNF-α—Tumor Necrosis Factor α; BDNF—Brain Derived Neurotrophic Factor.

**Table 3 ijms-24-03081-t003:** A Collection of clinical studies on the effects of probiotic supplementation and molecular mechanisms of its action in patients with major depression.

Author, Year, Country	Study Design	Characteristics of the Research Group	Sample Size (N), Sex, Average Age	Intervention	Study Methodology: Measurements of Effects, Duration of the Intervention	Conclusions	Molecular Mechanism of Action
Akkasheh et al., 2016, Iran [24]	Double-blind, randomized placebo-controlled trial	40 patients with a major depressive disorder	40, M, F, Average age 38 years old	*Lactobacillus acidophilus* (2 × 10^9^ CFU), *Lactobacillus casei* (2 × 10^9^ CFU), and *Bifidobacterium bifidum* (2 × 10^9^ CFU) 1 capsule per day	BDI, insulin, glutathione concentrations, hs-CRP, lipid profiles, total antioxidant capacity levels; Duration—8 weeks	Probiotic administration in patients with MDD for 8 weeks had beneficial effects on the Beck depression inventory, insulin, homeostasis model assessment of insulin resistance, hs-CRP concentrations, and glutathione concentrations, but did not influence fasting plasma glucose, the homeostatic model assessment of beta cell function, quantitative insulin sensitivity check index, lipid profiles, and total antioxidant capacity levels	The study was designed to determine the effects of probiotic intake on symptoms of depression and metabolic status in patients with major depressive disorder
Arifdjanova et al., 2021, Russia [18]	Open randomized clinical trial	119 depressed patients taking citalopram	119, M, F, Average age 32 years old	Bac-Set-Forte (*Lactobacillus casei PXN 37, Lactobacillus plantarum PXN 47, Lactobacillus rhamnosus PXN 54, Lactobacillus acidophilus PXN 35, Lactobacillus bulgaricus PXN 39, Lactobacillus helveticus PXN 45, Lactobacillus salivarius PXN 57, Lactobacillus fermentum PXN 44, Lactococcus lactis ssp. lactis PXN 63, Streptococcus thermophilus PXN 66,Bifidobacterium bifidum PXN 23, Bifidobacterium breve PXN 25, Bifidobacterium longum PXN 30,* *Bifidobacterium infantis PXN 27*) (2 × 10^9^ CFU) 1 capsule per day	HAMD-17, IL-6, TNF-α, NO; Duration—6 weeks	Significant improvement in HAMD-17 score in probiotic group after 6-weeks of treatment Patients undergoing probiotic therapy had a statistically significant decrease in levels of cortisol, dopamine, IL-6, TNF-α, and NO, as well as a more pronounced reduction in depression symptoms	Investigation of the probiotic therapy effect on the profile of the psycho-emotional state and nature of neuro-immune-endocrine changes by normalizing the concentration of glucocorticoids, catecholamines, proinflammatory cytokines, and nitric oxide in the patients undergoing combination therapy (antidepressant + probiotic) compared with patients receiving antidepressant-and-placebo therapy
Chen et al., 2021, Taiwan [25]	Open Trial	11 patients with major depressive disorder	11, M, F, Average age 39 years old	*Lactobacillus plantarum* PS128 (3 × 10^10^ CFU) 2 capsules per day	HAMD, TNF-α, zonulin, hs-CRP, IL-6, and I-FABP; Duration—8 weeks	There were decreased depressive symptoms as well as serum levels of high sensitivity c-reactive proteins, interleukin-6, and tumor necrosis factor-α, zonulin, and intestinal fatty acid binding protein, and the composition of gut microbiota did not significantly change after 8-week PS128 intervention	This study investigates the changes in biomarkers of gut permeability and gut microbiota after probiotic intervention in patients with MDD
Heidarzadeh-Rad et al., 2020, Turkey [20]	Double-blind, randomized placebo-controlled trial	110 depressed patients	110, M, F, Average age 36 years old	*Lactobacillus helveticus R0052* and *Bifidobacterium longum R0175* (10 × 10^9^ CFU) 1 sachet per day	BDI, BDNF; Duration—8 weeks	Improved depressive symptoms and significant decrease in BDI score in probiotic group The supplementation resulted in significantly higher serum BDNF levels	Investigating the influence of the gut microbiota on the brain-derived neurotrophic factor on antidepressant response in depressive patients
Kazemi et al., 2018, Iran [14,16]	Double-blind, randomized placebo-controlled trial	110 patients with a major depressive disorder	110, M, F, Average age 36 years old	Probiotic group: *Lactobacillus helveticus R0052* and *Bifidobacterium longum R0175* (10 × 10^9^ CFU) Prebiotic group: galactooligsaccharide 1 sachet per day	BDI; TNF-α, IL-1β, IL-6, IL-10, urinary cortisol levels; Duration—8 weeks	Significant decrease in BDI score in probiotic group after 8-weeks of treatment; For all patients, cytokine and cortisol levels were not statistically significant between the groups; The probiotic did not make a significant increase in the tryptophan to branch chain amino acids ratio compared to the placebo, while the prebiotic increased it significantly compared to the placebo	Investigating the correlation between branch chain amino acid concentrations and the severity of major depressive disorder and the potential of the gut microbiome to affect serum branch chain amino acid concentrations; Investigation of the effect of prebiotics and probiotics on serum inflammatory cytokines and urinary cortisol in patients with major depressive disorder
Reininghaus et al., 2020, Austria [22]	Double-blind, randomized placebo-controlled trial	82 depressed patients	82, M, F, Average age 43 years old	*Bifidobacterium bifidum* W23, *Bifidobacterium* *lactis* W51, *Bifidobacterium lactis* W52, *Lactobacillus acidophilus* W22, *Lactobacillus* *casei* W56, *Lactobacillus paracasei* W20, *Lactobacillus plantarum* W62, *Lactobacillus* *salivarius* W24, and *Lactobacillus lactis* W19 (7.5 × 10^12^ CFU), and 125 mg of D-Biotin (vitamin B7), 30 mg of common horsetail, 30 mg of fish collagen, and 30 mg of keratin	Microbiome samples were bioinformatically explored using QIIME; HAMD, BDI-II, SCL-90R, MSS, Vitamin B6 and B7; Duration—4 weeks	Probiotic intervention compared to placebo only differed in the microbial diversity profile, not in clinical outcome measures. *Ruminococcus gauvreauii* and *Coprococcus* 3 were more abundant, and diversity was higher in the probiotics group after 28 days. Upregulation of vitamin B6 and B7 synthesis which underlines the connection between the quality of diet, gut microbiota, and mental health through the regulation of metabolic functions	Examination of the composition of the gut microbiome and biomarkers linked to pathways regulating inflammation and metabolism. Determination of vitamin B6 and B7 synthesis levels as modulators of metabolic regulation in depressed individuals
Reiter et al., 2020, Austria [27]	Double-blind, randomized placebo-controlled trial	61 patients with major depressive disorder	61, M, F, Average 43 years old	OMNi-BiOTiC^®^ STRESS Repair 7.5 × 10^9^ CFU per bag (*Lactobacillus casei W56, Lactobacillus acidophilus W22, Lactobacillus paracasei W20, Bifidobacterium lactis W51, Lactobacillus salivarius W24, Lactococcus lactis W19, Bifidobacterium lactis W52, Lactobacillus plantarum W62, Bifidobacterium bifidum W23*)	HAMD, BDI-II, IL-6, TNF-α, NFKB1; Duration—4 weeks	No results showing improvement in patients’ mood after 4-week treatment; The findings indicate that patients with a four-week probiotic intake showed decreasing IL-6 gene expression levels while the placebo group showed increased gene expression levels of IL-6. This result confirms the positive effects of multispecies probiotics on mild inflammation in depressive disorders. However, on a univariate level no significant effects were found in TNF-α, nor NFKB1	RNA isolation from peripheral blood mononuclear cells (inflammation and endogenous control genes)
Rudzki et al. 2019, Poland [28]	Double-blind, randomized placebo-controlled trial	79 participants with major depressive disorder	79, M, F, Average 39 years old	*Lactobacillus plantarum 299v* (1 × 10^9^ CFU) 2 capsules per day	HAMD, TRP, KYN, KYNA, 3HKYN, AA, 3HAA, TNF-α, IL-6, IL-1β, cortisol plasma concentrations; Duration—8 weeks	No results showing an improvement in patients’ mood after 8-week treatment; Study presented a significant decrease in KYN concentration and 3HKYN:KYN ratio in the *LP299v* group compared to the placebo group. However, there were no significant changes in TNF-α, IL-6, IL-1β and cortisol concentrations in either probiotic nor placebo groups	Investigation of pharmacological and psychobiotical modulation of kynurenine pathway and modulation of proinflammatory cytokines and cortisol (characterizing HPA axis activity) levels by augmentation of probiotic bacteria *Lactobacillus Plantarum 299v* on cognitive, affective, and immune parameters of depressed patients undergoing SSRI treatment
Schaub et al., 2022, Switzerland [15]	Double-blind, randomized placebo-controlled trial	60 patients with current depressive episodes	60, M, F, Average age 39 years old	Vivomixx^®^ (*Streptococcus thermophilus NCIMB 30438, Bifidobacterium breve NCIMB* *30441, Bifidobacterium longum NCIMB 30435 (Re-classified as B. lactis), Bifidobacterium infantis NCIMB 30436 (Re-classified as B. lactis), Lactobacillus* *acidophilus NCIMB 30442, Lactobacillus plantarum NCIMB 30437, Lactobacillus paracasei NCIMB 30439, Lactobacillus delbrueckii subsp. Bulgaricus NCIM30440*) The daily dose contained 900 billion CFU/day	HAMD, Quantitative microbiome profiling and neuroimaging; Duration—4 weeks	Significant improvement in HAMD scores in probiotic group. Probiotics increased the abundance of the genus of the *Lactobacillus*, which was associated with decreased depressive symptoms in the probiotics group	Research focused on the effect of a short-term, high-dose probiotic add-on therapy on depressive symptoms by exploring gut microbiota composition as well as brain structure and function by rRNA data processing, enterotyping, and diversity measurements
Tian et al., 2022, China [29]	Double-blind, randomized placebo-controlled trial	45 patients with a major depressive disorder	45, M, F, Average age 51 years old	*Bifidobacterium breve CCFM1025* (1 × 10^10^ CFU) 1 sachet per day	HAMD, TNF-α, IL-1β, cortisol; Duration—4 weeks	Improved mood—significant decrease in HAMD-17 score in probiotic group; No statistical difference was observed in the concentration changes in serum cortisol, TNF-α, and IL-1β	The study was conducted in order to clarify the probiotics’ mechanisms of action. Among those mechanisms, research was focused on the gut microbiome and its tryptophan metabolism. Cortisol, TNF-α, and IL-1β in serum were measured. Additionally, serotonin turnover in the circulation, gut microbiome composition, and tryptophan level were evaluated

F—female; M– male; CFU—Colony Forming Unit; BDI—Beck Depression Inventory; MDD—major depressive disorder; hsCRP—High-Sensitivity CRP; HAMD-17—17-item Hamilton depression rating scale; IL-6—Interleukin-6; TNF-α—Tumor Necrosis Factor α; NO—nitric oxide; HAMD—Hamilton Depression Rating Scale; I-FABP—intestinal fatty-acid binding protein; BDNF—Brain Derived Neurotrophic Factor; IL-1β—Interleukin-1β; IL-10—Interleukin-10; QIIME—Quantitative Insights Into Microbial Ecology; BDI-II—Beck Depression Inventory-II; SCL-90R—Symptom Checklist-90-Revised; MSS—Mania Self Rating Scale; NFKB1—Nuclear Factor Kappa B Subunit 1; TRP—tryptophan; KYN—kynurenine; KYNA—kynurenic acid; 3HKYN—3-hydroxykynurenine; AA—anthranilic acid; 3HAA—3-hydroxy anthranilic acid.

**Table 4 ijms-24-03081-t004:** A collection of clinical studies on the effects of probiotic supplementation in depressed patients with somatic diseases.

Author, Year, Country	Study Design	Characteristics of the Research Group	Sample Size (N), Sex, Average Age	Intervention	Study Methodology: Measurements of Effects, Duration of the Intervention	Conclusions	Molecular Mechanism of Action
Haghighat et al., 2019, Iran [30]	Double-blind, randomized placebo-controlled trial	75 hemodialysis patients	75, M, F, Average age 46 years old	Probiotics (*Lactobacillus acidophilus T16, Bifidobacterium bifidum BIA-6, Bifidobacterium lactis BIA-7, and Bifidobacterium longum BIA-8*) (2.7 × 10^7^ CFU); synbiotics (5 g above mentioned probiotics+ 15 g of prebiotics)	HADS, BDNF; Duration—12 weeks	Synbiotic supplementation resulted in better results in HADS score than the probiotic and placebo group; The serum BDNF increased significantly in the synbiotic group compared to the placebo and probiotic groups. In the probiotic group, the supplementation did not result in a greater improvement in depression symptoms and serum BDNF level	Investigating the effect of and probiotic supplementation on serum brain-derived neurotrophic factor levels in hemodialysis patients
Majeed et al., 2018, India [26]	Double-blind, placebo controlled, randomized, multi-center, pilot clinical study	40 patients with a major depressive disorder and IBS	40, M, F, Average age 40 years old	*Bacillus coagulans MTCC 5856* (2 × 10^9^ CFU) 1 capsule per day	HAMD, MADRS, CES-D, serum myeloperoxidase; Duration—12 weeks	Significant improvement in HAMD, MADRS, and CES-D score, improvement in symptoms of irritable bowel disease; Serum myeloperoxi-dase as an inflammatory biomarker was significantly reduced	Measuring the effect of *Bacillus coagulans* MTCC 5856 on the serum myeloperoxidase which is responsible for the production of free radicals which leads to cellular oxidative stress and is linked to depression
Moludi et al., 2019, Iran [32]	Double-blind, randomized placebo-controlled trial	44 patients with a recent diagnosis of myocardial infarction	44, M, F, Average age 56 years old	*Lactobacillus rhamnosus* (1.6 × 10^9^ CFU) 1 capsule per day	BDI-II, QOLS, malondialdehyde levels, hs-CRP; Duration—12 weeks	Probiotic supplementation in patients with percutaneous coronary intervention post-myocardial infarction prompted an improvement in depressive symptoms and markers of oxidative stress and inflammation; The results of this trial show that probiotic supplementation resulted in decreased serum hs-CRP levels and a decrease in malondialdehyde levels	A study focusing on investigating the relationship of inflammation and oxidative stress onset and depression by determining the concentration of pro-inflammatory cytokines and oxidative stress biomarkers during probiotic administration
Moludi et al., 2021, Iran [31]	Double-blind, randomized placebo-controlled trial	96 patients with coronary artery disease	96, M, F, Average age 51 years old	Prebiotic group—one sachete containing 15 g inulin, Probiotic group- *Lactobacillus rhamnosus* (1.9 × 10^9^ CFU) 1 capsule per day, Co-supplemented group (both inulin and *Lactobacillus rhamnosus*)	BDI-II, IPAQ, QQLS, STAI-Y, Blood Pressure, levels of IL-10, TNF-α, lipid profile, hs-CRP, LPS, FBS, total cholesterol, LDL-C, TG, HDL-C; Duration—8 weeks	BDI and STAI-trait scores were significantly decreased in the probiotics group; no significant differences in serum concentrations of hs-CRP, LPS, TNF-α, IL-10, FBS, total cholesterol, LDL-C, TG, HDL-C, DBP, and SBP within groups	Study evaluating the effects of inulin and the *Lactobacillus rhamnosus* on inflammatory markers in patients with coronary artery disease
Raygan et al., 2018, Iran [17]	Double-blind, randomized placebo-controlled trial	60 diabetic patients with coronary artery disease	60, M, F, Average age 71 years old	*Lactocare^®^ (Lactobacillus casei, Lactobacillus acidophilus, Lactobacillus rhamnosus, Lactobacillus bulgaricus, Bifidobacterium breve, Bifidobacterium longum, Streptococcus thermophiles with prebiotic froctooligosaccharide)* (8 × 10^9^ CFU) 1 capsule per day + 50,000 IU vitamin D every 2 weeks	BDI, BAI, serum hs-CRP, plasma NO, glycemic control and HDL-c levels; Duration—12 weeks	Significant improvement in BDI and BAI scores, serum hs-CRP, plasma NO, glycemic control, and HDL-c levels among participants in vitamin D and probiotic co-supplementation compared to the placebo group	A study examining the interaction of antioxidant and anti-inflammatory effects of vitamin D and probiotics on mental health parameters, serum hs-CRP, plasma NO, TAC, glycemic control, and HDL-c levels
Raygan et al. 2019, Iran [19]	Double-blind, randomized placebo-controlled trial	54 diabetic patients with coronary artery disease	60, M, F, Average age 64 years old	*Lactocare^®^ (Lactobacillus casei, Lactobacillus acidophilus, Lactobacillus rhamnosus, Lactobacillus bulgaricus, Bifidobacterium breve, Bifidobacterium longum, Streptococcus thermophiles with prebiotic froctooligosaccharide)* (8 × 10^9^ CFU) 1 capsule per day + selenium 200 μg per day	BDI, BAI, PSQI, serum hs-CRP, plasma NO, glycemic control, and HDL-C; Duration—12 weeks	Significant improvement in BDI and BAI scores, serum hs-CRP, plasma NO, glycemic control and HDL-cholesterol levels in the probiotic and selenium co-supplementation group compared to the placebo group	A study examining the interaction antioxidant and anti-inflammatory effects of selenium and probiotics on mental health parameters, serum hs-CRP, plasma NO, TAC, glycemic control, and HDL-cholesterol levels
Zhang et al., 2021, China [33]	Double-blind, randomized placebo-controlled trial	82 patients with a major depressive disorder and constipation	82, M, F, Average age 45 years old	*Lacticaseibacillus paracasei YIT 9029* (1 × 10^10^ CFU) 1 capsule per day	BDI, HAM-D, fecal microbiome analysis, IL-1β, IL-6, TNF-α; Duration—9 weeks	Mood improvement in both research groups with no significant differences between groups. After 9 weeks of the probiotic intervention, the IL-1β, IL-6, and TNF-α were significantly decreased in both groups, the IL-6 levels were significantly lower in the probiotic group than in the placebo group	Investigation of the effect of *Lacticaseibacillus paracasei* strain Shirota (LcS) on constipation in patients with depression with specific etiology and gut microbiota and on depressive regimens by microbiota compositional analyses were performed on the V3–V4 region of the 16S rRNA gene and measured on the serum inflammatory factors: interleukins IL-1β, IL-6, and TNF-α

F—female; M—male; CFU—Colony Forming Unit; HADS—Hospital Anxiety and Depression Scale; BDNF—Brain Derived Neurotrophic Factor; HAMD—Hamilton Depression Rating Scale; MADRS—Montgomery and Asberg Depression Rating Scale; CES-D—Center for Epidemiologic Studies Depression Scale; BDI-II—Beck Depression Inventory-II; QOLS—The Quality of Life Scale; hsCRP—High-Sensitivity CRP; IPAQ—International Physical Activity Questionnaire; STAI-Y—Spielberger state-trait anxiety inventory form Y; IL-10—Interleukin-10; TNF-α—Tumor Necrosis Factor α; LPS—lipopolysaccharides; FBS—fasting blood sugar; LDL-C—low-density lipoprotein cholesterol; TG—triglyceride; HDL-C—high-density lipoprotein cholesterol; DBP—diastolic blood pressure; SBP—systolic blood pressure; BDI—Beck Depression Inventory; BAI—Beck Anxiety Inventory; TAC—total antioxidant capacity; NO—nitric oxide; PSQI—Pittsburgh Sleep Quality Index.

### 3.3. Effects of Probiotic Supplementation and Molecular Mechanisms of Its Action

#### 3.3.1. Effects of Probiotic Supplementation and Molecular Mechanisms of Its Action in Healthy Participants with Depressive or Anxiety Symptoms

A total of two clinical studies on healthy participants evaluating the effectiveness of probiotic supplementation on the occurrence of depressive symptoms and the molecular mechanisms of its action were identified (Table 2). They were assessed as “low risk of bias” [21] and “some concerns” [23] and provided contradictory results. Probiotic therapy significantly reduced depressive and anxiety symptoms and improved sleep quality in the probiotic group compared to the placebo group in the study conducted by Lee et al. [21], while the study undertaken by Romijin et al. [23] failed to demonstrate such an outcome. In both studies, the researchers were also focused on pro-inflammatory cytokines. The study by Lee et al. [21] confirmed the correlation between the relief of depressive symptoms in probiotic-treated patients with a reduction in levels of the pro-inflammatory cytokine (IL-6), while the other study failed to find such an association [23]. The latter study has also attempted to determine parameters such as BDNF and vitamin D in patients receiving probiotics and found no apparent correlations [23]. 

#### 3.3.2. Effects of Probiotic Supplementation and Molecular Mechanisms of Its Action in Patients with Major Depression

A total of 10 clinical studies evaluating the effectiveness of probiotic administration in patients with major depression and the molecular mechanisms of its action were identified (Table 3). Only one of these studies was assessed as having a “low risk of bias” [14], four had “some concerns” [15,20,22,28], while as many as 5, i.e., as much as half had a “high risk of bias” [18,24,25,27,29]. The majority of these studies on the efficacy of probiotics in patients with depression showed the effectiveness of such an intervention when considering both all the identified studies (7/10; 70%) and good-quality studies only (3/5, 60%) [14,15,20]. Six out of 10 studies measured parameters related to inflammation, which included TNF-α, IL-1β, IL-6, IL-10, and hs-CRP [14,18,25,27,28,29]. Only in two studies (33%) did the researchers note a significant decrease in inflammation biomarkers; however, these studies were assessed as having a “high risk of bias” [18,27]. None of the good-quality studies showed a change in inflammatory parameters in depressed patients treated with probiotics; thus, the interpretation of results related to molecular mechanisms of probiotics in depressed patients should be approached with great caution.

Another addressed molecule was the natural steroid hormone involved in HPA, namely cortisol [14,18,28,29]. Interestingly, only one study showed significant differences in the levels of this parameter in depressed patients treated with probiotics compared to the placebo (1/4; 25%). Nevertheless, after excluding low-quality studies, no study confirmed such a relationship. Subsequently, lipid profiles, a protein called the fatty acid binding protein, and amino acid KYN, which is a metabolite of tryptophan, and its ratio expressed as 3KHYN: KYN were described [24,25,28]. Among those studies, only Rudzki et al. found a significant decrease in KYN concentration and the 3HKYN: KYN ratio in the probiotic group [28]. It is worth noting that studies examining a lipid profile failed to demonstrate any relationship; however, they were considered to have a “high risk of bias” [24,25]. Interesting data were also provided by a study by Heidarzadeh-Rad et al. [20]. The authors found significantly higher serum BDNF levels in the probiotic group compared to the placebo in depressed patients. There was also a genetic approach. Schaub et al. performed rRNA data processing based on microbiome profiling after probiotic administration [15]. Probiotics increased the abundance of the genus of *Lactobacillus*, which was probably associated with decreased depressive symptoms in the probiotics group [15].

#### 3.3.3. Effects of Probiotic Supplementation in Depressed Patients with Somatic Diseases

A total of seven clinical studies evaluating the effectiveness of probiotic supplementation in depressed patients with chronic diseases were identified (Table 4). Two of these studies were assessed as having a “low risk of bias” [17,19], four as “some concerns” [30,31,32,33], and one as a “high risk of bias” [26]. The vast majority of studies showed an improvement in the mood of probiotic-treated patients (6/7; 86%). Five out of seven studies described immunological profiles based on the measurement levels of hs-CRP/CRP [17,19,31,32], IL-10 with TNF-α [31], and IL-6 [33]. The result of the studies focused on immunological biomarkers and showed that probiotic supplementation resulted in decreased serum hs-CRP/CRP levels in 75% of studies (3 out of 4) [17,19,32], and in IL-6 levels in 100% of studies (1/1) [33]. However, there were no significant differences in serum concentrations of hs-CRP, IL-6, and TNF-α in the study conducted by Moludi et al. [31]. Parameters related to the central nervous system were described in three studies—the levels of NO [17,19] and BDNF [30] were measured. The study by Haghighat et al. found higher BDNF levels in hemodialysis-depressed patients treated with probiotics compared to the placebo group [30]. Moreover, three studies alluded to a lipid profile and measured for LDL cholesterol, as well as HDL cholesterol, lipopolysaccharides, and fasting blood sugar [17,19,31]. Two studies conducted by Raygan et al. found a significant improvement in plasma NO, glycemic control, and HDL-cholesterol levels [17,19]. In contrast, the study performed by Moludi et al. did not detect any significant changes in the lipid profile [31]. In addition, two studies were performed on markers of oxidative stress and confirmed the involvement of this group of parameters in reducing depressive symptoms in patients treated with probiotics [26,32]. This group of parameters includes serum myeloperoxidase [26] and malondialdehyde, which is a primary indicator of lipid peroxidation [32]. 

All outcomes concerning the molecular mechanisms of action for probiotics in all three groups are summarized in Table 5.

### 3.4. Meta-Analysis of Randomized Control Trials on the Effect of Probiotic Treatment on the Molecular Mechanism in Depressed Patients in Intervention (Probiotics) and Placebo (Control) Groups

The meta-analysis was performed using random effects models. Only parameters assessed in at least two good-quality studies (i.e., with low or moderate risk of bias) and with sufficient data regarding the mean differences and standard deviation (SD) in a given markers’ level for both the intervention (probiotic) and control (placebo) groups were considered. Consequently, the following parameters were analyzed: IL-1β [14,28,33], IL-6 [21,28,33], IL-10 [14,31], TNF-α [14,21,31], CRP [17,19,31,32], BDNF [20,23,30], cortisol [14,21,28], nitric oxide (NO) [17,19]. The mean difference and 95% confidence intervals (CI) for the effect of probiotics on serum levels of the respective markers in the intervention and control groups were estimated. Results from the random effects model showed that CRP levels in the probiotic group were significantly lower than in the placebo group (SMD = −0.47, 95% CI [−0.75, −0.19], *p* = 0.001; Figure 4). As regards the BDNF and NO levels, in the probiotic group, these parameters were significantly higher than in the placebo group (SMD = 0.37, 95% CI [0.07, 0.68], *p* = 0.02; SMD = 0.97, 95% CI [0.58, 1.36], *p* < 0.0001, respectively, Figure 5 and Figure 6). There were no significant differences in IL-1β, IL-6, IL-10, TNF-α, and cortisol levels after probiotic administration between the intervention and control groups (IL-1β: SMD = 0.56, 95% CI [−0.14, 1.26], *p* = 0.11; IL-6: SMD = −0.21, 95%CI [−0.63, 0.21], *p* = 0.32; IL-10: SMD = 0.20, 95% CI [−0.20, 0.60], *p* = 0.33; TNF-α: SMD = −0.19, 95% CI [−0.45, 0.08], *p* = 0.16; cortisol: SMD = −0.10, 95% CI [−0.36, 0.15], *p* = 0.43; Figure 7, Figure 8, Figure 9, Figure 10 and Figure 11).

### 3.5. Discussion

This systematic review investigates the available studies on the molecular mechanisms in the use of probiotics in healthy adults with subclinical symptoms of depression or anxiety and in depressed patients with or without concomitant somatic disorders. All the included studies investigated immunological parameters, with an emphasis on pro-inflammatory cytokines. In addition, there were references to the study of parameters in relation to BDNF and NO levels, vitamin levels, amino acids, or lipid profiles, including vitamins B and D, HDL, and LDL cholesterol, fatty acid binding proteins, as well as tryptophan metabolites and oxidative stress markers. 

In recent years, interest in the interplay between the brain and gut microbiota has continued to grow. This relationship has been referred to as the gut-microbiota-brain axis (MGBA) [37]. This connection can cause changes in the functioning of the central nervous system, the gastrointestinal tract, or human behavior [38]. While focusing on the communication of the gut microbiota with the central nervous system, neuronal, immunological, and endocrine signaling were distinguished. When the MGBA functions properly, we speak of homeostasis, whereas if it is disturbed, we can expect dysfunctions in the body, including depression [39].

In this review, we found evidence supporting the positive effect of probiotic administration in depressed patients, in particular among depressed patients with co-morbid chronic diseases. Interestingly, the results for the group of patients with depression and various concomitant somatic conditions were more consistent and provided strong evidence for the antidepressant effectiveness of probiotics in this population compared to the group with depression alone. The reported effectiveness of probiotic therapy in depressed patients with and without coexisting somatic conditions was 86% and 60%, respectively. The differences in the effect of probiotics between studies may be due to a number of factors. The possible reasons for these differences are the different timing of probiotic administration, as well as the use of different strains of bacteria, and the use of probiotics or synbiotics. In the studies involving patients with a major depressive disorder, no improvement was observed in three studies [22,27,28]. However, in two of these studies [22,27], the administration of probiotics lasted only 4 weeks, whereas in most other studies it was at least 8 weeks. It is also noteworthy that the probiotic administration in the group of patients with depression without comorbid somatic illnesses lasted for a shorter period of time (range: 4–8 weeks), while in the group of patients with depression and somatic disorders, probiotics were used longer (range: 8–12 weeks). Therefore, it appears that the length of probiotic use may be a contributing factor to the effectiveness of such treatment. On the other hand, Rudzki et al. [28] were the only ones to use the *Lactobacillus plantarum 299v* strain in their study, finding no antidepressant efficacy in probiotics. Following the results from the studies related to patients with coexisting somatic conditions, this group was characterized by the greatest heterogeneity due to the different somatic diseases present among the patients. In general, different molecular processes are responsible for the course of different somatic diseases, and therefore, the administration of probiotic strains may result in different therapeutic effects in individual conditions. It should also be emphasized that differences in the effectiveness of probiotic therapy may also result from the use of different bacterial strains. Single-strain and multi-strain probiotics were used in the identified studies. The best results were achieved in studies where a two-strain mixture was used, consisting of *Lactobacillus helveticus R0052* and *Bifidobacterium longum R0175* [14,16,20,23], and Lactocare^®^ (*Lactobacillus casei, Lactobacillus acidophilus, Lactobacillus rhamnosus, Lactobacillus bulgaricus, Bifidobacterium breve, Bifidobacterium longum, Streptococcus thermophiles*) [17,19]. Other studies have used probiotics which are a three-strain mixture consisting of *Lactobacillus acidophilus, Lactobacillus casei*, and *Bifidobacterium bifidum*, a two-strain one using *Lactobacillus reuteri NK33* and *Bifidobacterium adolescentis NK98* and single-strain probiotics, including *Bacillus coagulans MTCC 5856*, *Bifidobacterium breve CCFM1025*, *Lacticaseibacillus paracasei YIT 9029, Lactobacillus rhamnosus.* Furthermore, one study considered the use of a combination of probiotics and prebiotics, known as synbiotics [30]. This study showed significant improvements in the HADS score and serum BDNF level only in the synbiotic group [30]. This may be due to the composition of the synbiotics, which consists of prebiotics and probiotics. Such a combination may additionally stimulate the growth and activity of the gut microbiota [30].

In contrast, findings on the efficacy of probiotics in a group of healthy participants with subclinical symptoms of depression/anxiety are inconclusive. Two of the identified studies provided conflicting results. In this case, the discrepancies in the results may be the result of the different strains of probiotics administered in the trials as well as the different doses.

The various effects between the group of patients with depression solely and with depression and a concomitant somatic condition were particularly pronounced in the aspect which was the main focus of this review—the connection of molecular mechanisms with the effectiveness of probiotic treatment. One of the most important foundations of gut microbiota homeostasis is its relationship with immune reactivity. TNF-α and IFN-γ production is associated with specific metabolic pathways of the microbiota residing in the gut: palmitoleic acid metabolism and the degradation of tryptophan to tryptophol, which is a tryptophan metabolite, has been shown to have a strong inhibitory effect on the TNF-α response. In turn, palmitoleic acid inhibits apoptosis which is induced by a combination of IL-1β and IFN-γ. In the case of inflammation, patients have higher levels of pro-inflammatory T-helper cells and pro-inflammatory cytokines such as IL-1β, IL-6, IL-10, and TNF-α, as well as higher CRP levels [17,19,21,32,33]. The concentration of CRP increases within hours of infection or after tissue damage and acts as a very useful indicator for monitoring inflammation [40,41]. In this review, we found consistent evidence for an association between two inflammatory parameters—hs-CRP and IL-6 in depressed patients with concomitant somatic disorders [17,19,31,32], as confirmed in the meta-analysis (SMD = −0.47, 95% CI [−0.75, −0.19], *p* = 0.001), whereas no conclusive evidence of such an association was found in the group of patients with depression alone, as well as in the group of healthy people with subclinical symptoms of depression. However, it is worth mentioning that in the latter group, reduced levels of the cytokine IL-6 were observed in one of the two available studies evaluating this parameter and in patients with subclinical symptoms of depression, anxiety, and insomnia (thus, in 50% of the research to date) [21,23]. Establishing the association of these parameters in a healthy population with subclinical depressive or anxiety symptoms would therefore be highly desirable. This could potentially be an important element in the prevention of depressive and anxiety disorders if probiotics prove to be effective in this population. 

We hypothesized that the observed discrepancies in terms of the effects on inflammatory parameters during probiotic treatment might be due to differences in the study populations in terms of, among other things, the severity of depressive symptoms, inflammation process ongoing in the organism (noted in somatic diseases), as well as differences in the mean age and sex ratio of the study samples. Regarding inflammatory markers, this could be supported by the observation that changes in markers were more pronounced in the group of depressed patients with somatic illness than in the group with depression alone. The function of the gut-associated lymphoid tissue (GALT), the aforementioned immune system derived from the intestinal mucosa, may be influenced by genetic factors and host-environment interactions while also being influenced by age and gender [42]. Changes in these immune responses are primarily cytokine-dependent. In elderly individuals, there is a marked impairment in the production of cytokines, T-helper, IL-22, and IFN-γ. Moreover, differential susceptibility between women and men has been reported in terms of susceptibility to infectious, autoimmune, and inflammatory diseases [43]. Indeed, women have been found to have a higher production of IL-17, and this observation may be the reason for the higher incidence of many autoimmune diseases, such as multiple sclerosis and rheumatoid arthritis in women. The observation of immune responses under varying environments is also an issue worthy of attention. The rationale for these variable responses in different months of the year is likely due to the seasonality of infectious diseases and the various metabolites released by microbiota-diet interactions when a seasonal diet is implemented [42]. It should also be emphasized that differences in the effectiveness of probiotic therapy may also result from the use of different bacterial strains. Single-strain and multi-strain probiotics were used in identified studies. The best results were achieved in studies where a two-strain mixture was used, consisting of *Lactobacillus helveticus R0052* and *Bifidobacterium longum R0175* [14,16,20,23], and Lactocare^®^ (*Lactobacillus casei, Lactobacillus acidophilus, Lactobacillus rhamnosus, Lactobacillus bulgaricus, Bifidobacterium breve, Bifidobacterium longum, Streptococcus thermophiles*) [17,19]. Other studies have used probiotics which are a three-strain mixture consisting of a *Lactobacillus acidophilus, Lactobacillus casei*, and *Bifidobacterium bifidum*, a two-strain one using *Lactobacillus reuteri NK33* and *Bifidobacterium adolescentis NK98* and single-strain probiotics, which included *Bacillus coagulans MTCC 5856*, *Bifidobacterium breve CCFM1025*, *Lacticaseibacillus paracasei YIT 9029, Lactobacillus rhamnosus.* Furthermore, one study considered the use of a combination of probiotics and prebiotics, which are known as synbiotics [30].

A further subject worth discussing is the effects of probiotics on hypothalamic-pituitary-adrenal (HPA)-related brain parameters. One of the proposed mechanisms of action for probiotics was a link to the reduction in the stress hormone cortisol. Chronic stress is similarly a factor that leads to depression [14]. Moreover, Mikocka-Walus et al. have found that the occurrence of depression often goes hand in hand with disturbances in gut physiology [44]. This process might be related to the HPA axis [41]. Sudo et al. demonstrated that the administration of probiotic strains of *Bifidobacterium infantis* regulates the dysfunctional HPA axis, therefore restoring balance [45]. Such reports support the theory that the gut microbiota has modulatory capabilities which may influence the HPA axis and stress response. In this review, we identified three studies in which the authors assessed changes in cortisol levels with probiotic therapy in depressed patients [14,28,29]. However, none of the above studies confirmed such an association. Therefore, this topic remains unresolved at this point; a more in-depth investigation is warranted.

On the contrary, in the case of the brain-derived neurotrophic factor (BDNF), we found evidence of an association of the level of this parameter with the resolution of depressive symptoms in probiotic-treated patients (SMD = 0.37, 95% CI [0.07, 0.68], *p* = 0.02). This applies both to patients with depression without as well as with coexisting somatic conditions. There was an increase in BDNF levels in two identified studies evaluating this parameter [20,30]. One of the concepts of this connection is linked to short-chain fatty acids (SCFAs), which are produced by the gut microbiota and may alter the expression of BDNF. Significantly lower levels are observed in depressed individuals compared to healthy people. When SCFAs levels are inappropriate, the integrity of the intestinal barrier is at risk of being compromised, resulting in the translocation of intestinal bacteria and their antigens, and such an event can cause mild inflammation in the host [46,47,48]. Some pharmacological treatments for depression, including ketamine, affect the mammalian target of rapamycin (mTOR), which functions as a serine/threonine protein kinase. Such an influence can modify mTOR signaling, thereby increasing BDNF activity and leading to an improvement in mood [49]. Another chemical compound worth mentioning is NO (nitric oxide): a simple molecule with very specific biological functions. Since it is dependent on environmental conditions, NO is able to mediate neuroprotection or neurotoxicity. The stabilization of NO levels was also correlated with an improvement in well-being [17,18,19,50]. Furthermore, NO is a part that forms trimethylamine N-oxide (TMAO). This compound is formed from trimethylamine (TMA), a metabolic product synthesized by the gut microbiota from choline and L-carnitine. The amount of choline and L-carnitine and, consequently, the quality of trimethylamine synthesis is dependent on the composition and homeostasis of the gut microbiota environment. Cases of patients with an abnormal gut microbiota composition and excessively high plasma trimethylamine N-oxide concentrations have been reported. The above-mentioned situation is undesirable due to the role of TMAO in the pathogenesis of many chronic diseases, such as atherosclerosis [51,52]. Maintaining normal TMAO levels is the provision of adequate amounts of choline and L-carnitine with the diet [51]. A factor that can disrupt homeostasis in TMA metabolism is the use of antibiotic therapy. Due to antibiotic intake, especially if it is a long-term process, dysbiosis of the gut microbiome and an increased number of undesirable bacteria are often observed, possibly affecting the increased production of TMA [53]. We found two studies that confirmed the stabilization of NO in probiotic-treated patients compared to the controls (SMD = 0.97, 95% CI [0.58, 1.36], *p* < 0.0001). However, this was confirmed only in two studies on depressed diabetic patients with coronary artery disease receiving probiotics [17,19]. 

Finally, an interesting phenomenon, although not yet fully understood, is the link between oxidative stress markers and the composition of the gut microbiota. Moreover, oxidative stress is related to various human diseases, which covers depression. We found two good-quality studies confirming the connection of probiotic therapy in depressed individuals with oxidative stress markers [26,32]. This finding was only confirmed in the group of depressed patients with a concomitant somatic condition, specifically with IBS [26], and with a recent diagnosis of myocardial infarction [32]. They concerned serum myeloperoxidase [26] and the serum concentration of malonaldehyde (MDA)—a biomarker of lipid peroxidation reflecting oxidative stress [32]. The levels of MDA decreased after probiotic therapy. Interestingly, malondialdehyde reacts with deoxyadenosine and deoxyguanosine in the DNA, forming DNA adducts, which may be mutagenic [32]. These findings were justified by the fact that recent studies have described the link between decreased antioxidant levels of glutathione and increased anhedonia severity which could probably lead to the appearance of neuroinflammation and oxidative stress in major depressive disorder. 

Lastly, it is worth pointing out the safety of probiotic therapy. When analyzing the safety of probiotic therapy in the included studies, no proved serious adverse effects were noted. 

### 3.6. Limitations

The studies described in this review represent a small group in quantitative terms, and thus, it is possible that the conclusions could change due to the addition of a large number of studies in the future. In addition, most studies have focused on evaluating the efficacy of probiotics in the treatment of depression, often considering only the effect on immune parameters as secondary outcomes. This may have influenced the inadequate estimation of the sample size that is required to detect significant differences in immunological parameters between groups. A shortage of research and highly heterogeneous outcome measurement tools in terms of molecular mechanisms precluded the meta-analysis of subgroups. Finally, most of the identified studies were assessed as moderate or even as a high risk of bias (in particular in the group of patients with major depression without somatic diseases), limiting the ability to draw firm conclusions. Furthermore, we included publications written only in the English language.

## 4. Conclusions

This systematic review supports the use of probiotics in depressed patients with or without concomitant somatic disorders. We also found consistent evidence of an increase in BDNF levels during probiotic treatment and the resolution of depressive symptoms in the above two populations of depressed patients. In contrast, in the case of inflammatory markers (CRP), oxidative stress markers, and NO levels, such a correlation was only confirmed in the group of depressed patients with somatic co-morbidities.

As there is a lack of large, good-quality RCTs on depressed patients receiving probiotics, future studies investigating the efficacy of specific doses and bacterial strains, as well as their specific interactions on given parameters, are needed. Furthermore, probiotic therapy, which goes hand in hand with antidepressants, might be more effective. As probiotic therapy appears to be safe in different populations and somatic conditions, it could be a promising option as an add-on treatment. Therefore, RCTs with probiotics in depressed patients, especially of medium long-term duration (at least 8 weeks), would be much needed. As the identified studies only assessed patient populations with only a few somatic conditions (irritable bowel syndrome, constipation, myocardial infarction, coronary artery disease, and hemodialysis patients), studies on patients with depression and other somatic conditions are required.

In addition, most studies have focused on the determination of pro-inflammatory parameters, while single studies have noted that other relevant parameters such as, among others, parameters related to the lipid profile, central nervous system, reflecting levels of micro- and macromolecules, and markers of oxidative stress, also changed with probiotic intake, resulting in positive outcomes, which could also be investigated in more depth in the future.

Going hand in hand with scientific developments, it is necessary to analyze the latest reports on the molecular aspects of somatic diseases, as well as depression, and search for new molecular parameters that are active in the disease entity in question, followed by trials involving the administration of probiotics and measuring their effects on specific parameters.

Firm conclusions on the prophylactic efficiency of probiotics in the healthy population (only with subclinical depressive or anxiety symptoms) cannot be drawn due to the divergence of results on existing studies and the paucity of such studies. The same applies to the possible association of inflammatory markers with probiotic therapy in this population. Establishing the association of these parameters in a healthy population with subclinical depressive or anxiety symptoms would therefore be highly desirable. In particular, prospective long-term studies in a healthy population or with subclinical symptoms of depression or anxiety could be of great importance. This could potentially be an important element in the prevention of depressive and anxiety disorders if probiotics prove to be effective in this population.

## Figures and Tables

**Figure 1 ijms-24-03081-f001:**
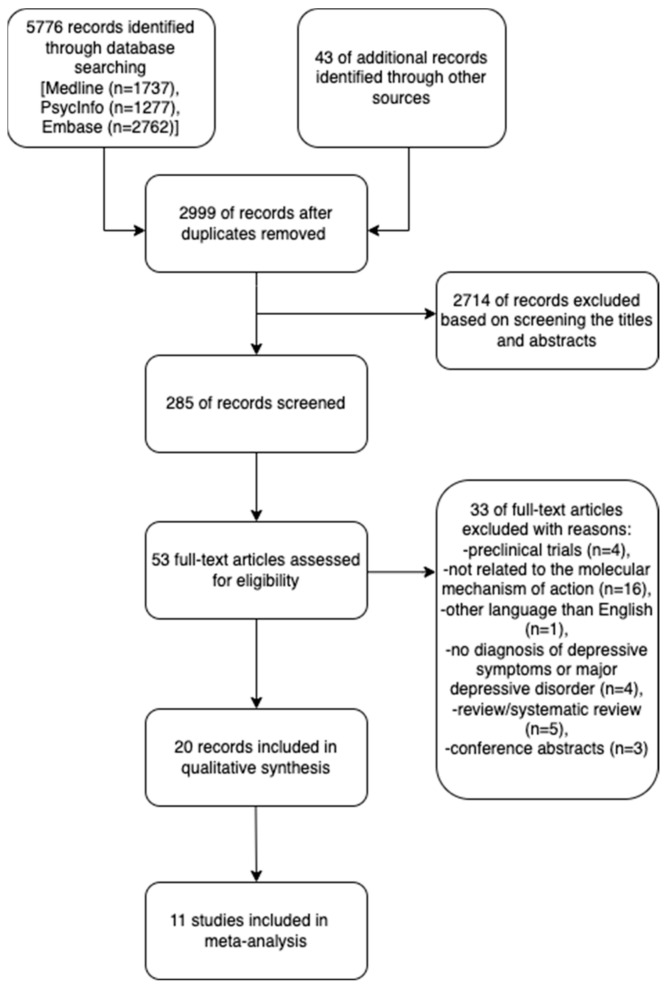
Prisma flow diagram [13] for clinical trials focusing on the molecular mechanism of action during the administration of probiotics in depressed patients included in the systematic review.

**Figure 2 ijms-24-03081-f002:**
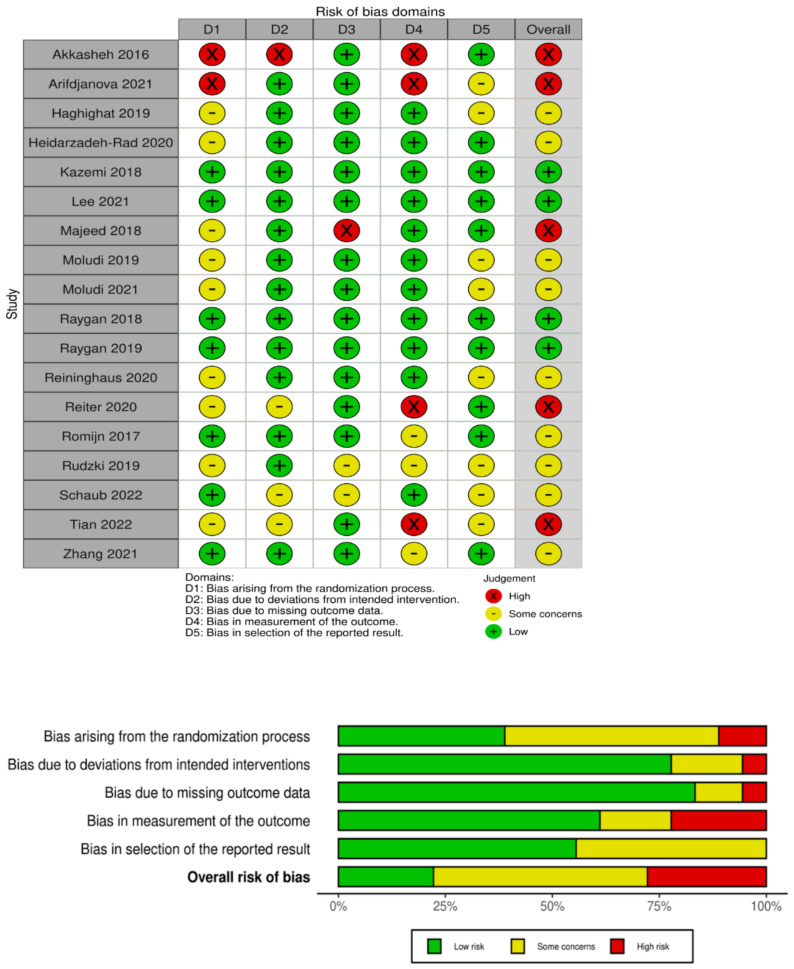
Risk of bias for interventional randomized studies separately and after summarizing with Rob2 tool [14,15,17,18,19,20,21,22,23,24,26,27,28,29,30,31,32,33,34,35].

**Figure 3 ijms-24-03081-f003:**
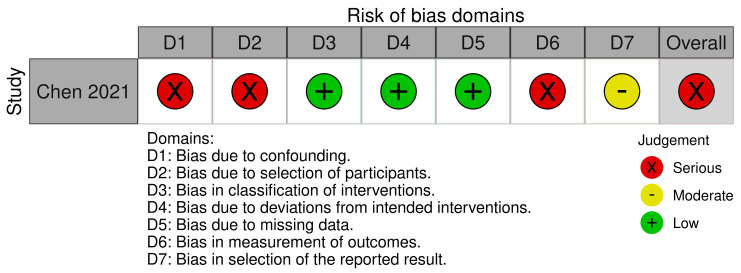
Risk of bias for interventional non-randomized study with ROBINS-I tool [25,36].

**Figure 4 ijms-24-03081-f004:**
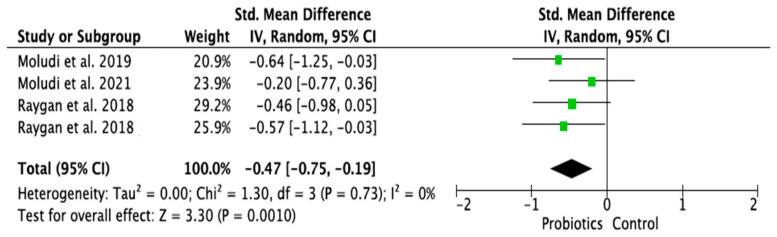
Forest plot for the effect of probiotic treatment on CRP level in probiotics and control groups [17,19,31,32]. Note: The green squares are the SMD for the post-hoc studies, the black rhombus is the graphical summary for all studies above.

**Figure 5 ijms-24-03081-f005:**
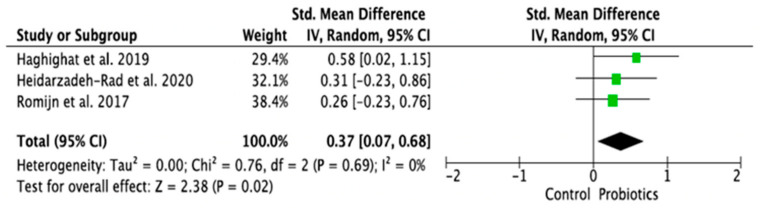
Forest plot for the effect of probiotic treatment on BDNF level in probiotics and control groups [20,23,30]. Note: The green squares are the SMD for the post-hoc studies, the black rhombus is the graphical summary for all studies above.

**Figure 6 ijms-24-03081-f006:**
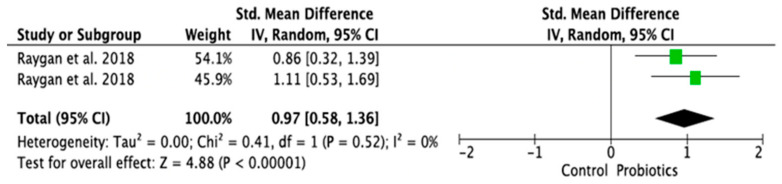
Forest plot for the effect of probiotic treatment on nitric oxide (NO) levels in probiotics and control groups [17,19]. Note: The green squares are the SMD for the post-hoc studies, the black rhombus is the graphical summary for all studies above.

**Figure 7 ijms-24-03081-f007:**
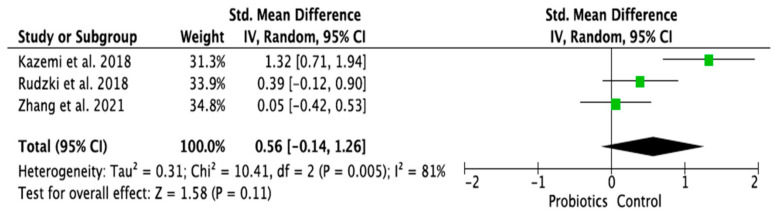
Forest plot for the effect of probiotic treatment on IFN-beta level in probiotics and control groups [14,28,33]. Note: The green squares are the SMD for the post-hoc studies, the black rhombus is the graphical summary for all studies above.

**Figure 8 ijms-24-03081-f008:**
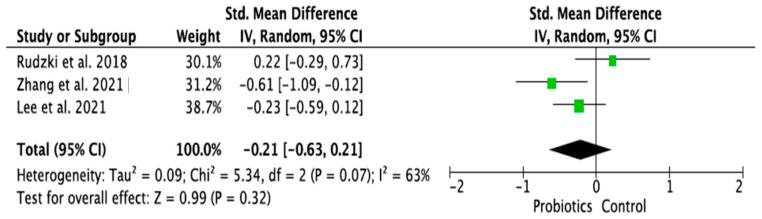
Forest plot for the effect of probiotic treatment on IL-6 level in probiotics and control groups [21,28,33]. Note: The green squares are the SMD for the post-hoc studies, the black rhombus is the graphical summary for all studies above.

**Figure 9 ijms-24-03081-f009:**
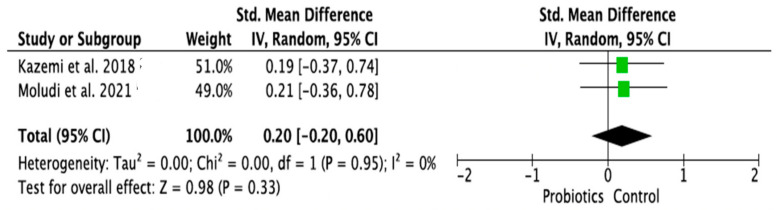
Forest plot for the effect of probiotic treatment on IL-10 level in probiotics and control groups [14,31]. Note: The green squares are the SMD for the post-hoc studies, the black rhombus is the graphical summary for all studies above.

**Figure 10 ijms-24-03081-f010:**
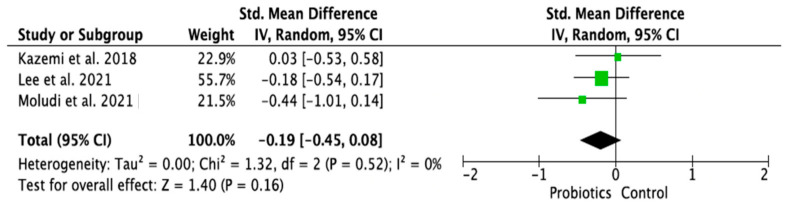
Forest plot for the effect of probiotic treatment on TNF-α level in probiotics and control groups [14,21,31]. Note: The green squares are the SMD for the post-hoc studies, the black rhombus is the graphical summary for all studies above.

**Figure 11 ijms-24-03081-f011:**
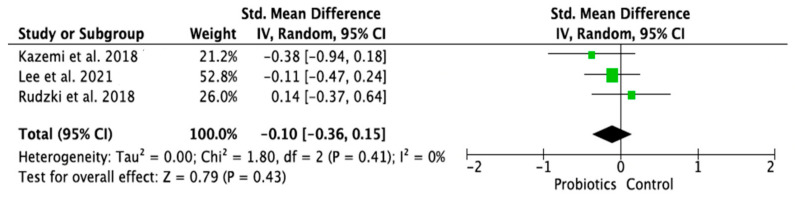
Forest plot for the effect of probiotic treatment on cortisol levels in probiotics and control groups [14,21,28]. Note: The green squares are the SMD for the post-hoc studies, the black rhombus is the graphical summary for all studies above.

**Table 1 ijms-24-03081-t001:** PICO questions.

	Clinical Studies
Patients	Individuals with depressive symptoms: major depression solely or comorbid with somatic illnesses
Intervention	Use of probiotics alone or as an add-on treatment
Comparison	No use of probiotics; placebo
Outcome	Evaluated molecular mechanism of action for probiotics

**Table 5 ijms-24-03081-t005:** Molecular mechanisms assessed in included studies.

Molecular Mechanism	Studies Reporting on Given Variable % (n/N)—Percent, (n—Number of Studies Where Variable Was Significant/N—Number of All Studies Evaluating Given Variable)		
	Healthy Population (with Subclinical Depressive/Anxiety Symptoms)	Depressed Patient without Comorbid Somatic Disorder	Depressed Patient with Comorbid Somatic Disorder
**Interleukin 6 (IL-6)** [14,21,23,28,33]	50% (1/2)	0% (0/2)	100% (1/1)
**Interleukin 1 beta (IL-1β)** [14,23,28,33]	0% (0/1)	0% (0/2)	0% (0/1)
**Interleukin 10 (IL-10)** [14,31]	-	0% (0/1)	0% (0/1)
**Tumor necrosis factor-alpha (TNF-α)** [14,23,28,31,33]	0% (0/1)	0% (0/2)	0% (0/2)
**C-reactive protein (CRP)** [17,19,23,31,32]	0% (0/1)	-	75% (3/4)
**Cortisol** [14,21,28]	0% (0/1)	0% (0/2)	-
**Brain-derived Neurotrophic Factor (BDNF)** [20,23,30]	0% (0/1)	100% (1/1)	100% (1/1)
**Nitric oxide (NO)** [17,19]	-	-	100% (2/2)
**Biomarkers of oxidative stress** [32]	-	-	100% (1/1)
**Lipids** [17,19,31]	-	-	67% (2/3)

Note: Outcomes after excluding studies assessed as high risk of bias.

## Data Availability

The data presented in this study are available on request from the corresponding author. The data are not publicly available due to privacy or ethical concerns.

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
