# Peer review of "Probiotics as a Tool for Regulating Molecular Mechanisms in Depression: A Systematic Review and Meta-Analysis of Randomized Clinical Trials"

_ijms, 2023, doi:10.3390/ijms24043081_

Round 1
Reviewer 1 Report
The article focuses on a topic of great interest due to the holistic thinking of researchers. However, the review fails to observe what it proposes. The selection of articles is already low. With the intention of getting the newest articles they restrict and 7 years (2015-current) when they could do a bigger prospect. In view of the article number reached.
Author Response
Thank you for this comment. As molecular medicine has advanced in recent years due to technological developments, we believe that seven years is a considerable period for recent reports. The number of studies included in our systematic review is closely related to the still small number of clinical trials conducted that focus on the molecular aspects of probiotics.
Thank you for reviewing our manuscript. We appreciate your time.
Reviewer 2 Report
I congratulate the authors for their excellent PRISMA systematic review. My only quibble is about the lack of a meta-analytic extension of it. Even an exploratory MA of a handful of trials would improve the overall quality of the report allowing for a more quantitative-based discussion, ideally promoting high-quality (ROBIN-1) studies over other findings. Thank you.
Author Response
Thank you for your important comment. As suggested by the reviewer, we have included an exploratory meta-analysis of good quality studies with sufficient data on the markers in concern. Relevant information can be found in the Methods, Results and Discussion sections. We have also modified the title accordingly. However, as the meta-analysis is not very extensive, we are not sure whether the title change is justified. We would appreciate your feedback on this issue.
Thank you very much for your insightful evaluation and valuable comment, we appreciate your time.
Reviewer 3 Report
This is an interesting review that consider the effects of probiotics in depression with a focus on mechanisms. In my opinion it is well written and easily readable. Some suggestion to improve the work. The authors could better discuss the discrepancies in the results obtainted in trials, analyzing above all the starting situation of patients and the methods of administration used. The authors could also expand Conclusions trying to propose, on the basis of their knowledge and of the literature analyzed, strategies aimed at improving depressive symptoms in different conditions and defining some aspects that future research should address.
Author Response
Thank you for your important comments. As suggested, we have discussed the discrepancies in the results obtained by analyzing the possible causes in more detail to make them clearer and more understandable to the reader. We have also improved the conclusions as suggested by the reviewer.
Thank you for reviewing our manuscript and for your valuable remarks. We appreciate your time.
Round 2
Reviewer 1 Report
Dear authors, the inclusion of the meta-analysis generated a significant improvement in the article, and the change of title improves when trying to understand the objective. Indeed, this is an open area of ​​study. And this article can shed light on that process.
Author Response
Thank you for your comment and kind review. We appreciate your time.